# Health-Promoting Quality of Life at Work during the COVID-19 Pandemic: A 12-Month Longitudinal Study on the Work-Related Sense of Coherence in Acute Care Healthcare Professionals

**DOI:** 10.3390/ijerph19106053

**Published:** 2022-05-16

**Authors:** Joana Berger-Estilita, Sandra Abegglen, Nadja Hornburg, Robert Greif, Alexander Fuchs

**Affiliations:** 1Institute for Medical Education, University of Bern, 3012 Bern, Switzerland; 2CINTESIS—Centre for Health Technology and Services Research, Faculty of Medicine, University of Porto, 4200-450 Porto, Portugal; 3Department of Health Psychology and Behavioural Medicine, University of Bern, 3012 Bern, Switzerland; sandra.abegglen@psy.unibe.ch (S.A.); nadja.hornburg@unibe.ch (N.H.); 4Department of Anaesthesiology and Pain Medicine, Inselspital, Bern University Hospital, University of Bern, 3010 Bern, Switzerland; robert.greif@insel.ch (R.G.); alexander.fuchs@insel.ch (A.F.); 5School of Medicine, Sigmund Freud University Vienna, 1020 Vienna, Austria

**Keywords:** COVID-19, acute care, work-related sense of coherence, healthcare workers, mental health

## Abstract

(1) Background: During a pandemic, the wellbeing of healthcare professionals is crucial. We investigated the long-term association of the Work-related Sense of Coherence (Work-SoC) and the evolution of psychological health symptoms of acute care healthcare professionals during the first year of the COVID-19 pandemic. (2) Methods: This longitudinal observational study enrolled 520 multinational healthcare professionals, who completed an online survey every three months from April 2020 to April 2021. Mixed linear models examined the associations between Work-SOC and COVID-19-related anxiety, perceived vulnerability, depressiveness, and psychological trauma symptomatology. (3) Results: Healthcare professionals with a higher Work-SoC reported lower levels of COVID-19-related anxiety, perceived vulnerability, depressiveness, and psychological trauma symptomatology in April 2020 than healthcare professionals with an average or lower Work-SoC, but the levels increased to higher values in April 2021. Healthcare professionals with a lower Work-SoC reported higher levels of depressiveness and psychological trauma symptomatology in April 2020 but lower levels in April 2021. (4) Conclusions: Healthcare professionals with higher levels of Work-related Sense of Coherence might be protected against variations in psychological symptoms for about three months, but this protection seems to decrease as the pandemic continues, resulting in mental health deterioration. In contrast, healthcare professionals with a lower Work-SoC might be protected at later stages of the pandemic.

## 1. Introduction

The ongoing coronavirus disease 2019 (COVID-19) pandemic has had a dramatic impact on healthcare professionals (HCPs) worldwide. Anaesthesia, critical care, and emergency room staff were exposed to extraordinary work-related challenges besides known stressors (e.g., high workload, long working hours, time pressure) [1]. Early in the COVID-19 pandemic, these HCPs were exposed to additional pandemic-related factors such as a high infection risk, scarce personal protection equipment, limited resources, many deaths, self-isolation, and stigmatisation [2,3,4,5]. During subsequent pandemic waves, although some shortages had been resolved and therapeutic protocols had been established, psychological problems related to the ongoing epidemic surfaced among HCPs [6].

Since the COVID-19 epidemic emerged, several publications have raised concerns that front-line HCPs who have direct contact with infected patients [7] are suffering from post-traumatic stress, anxiety, depressiveness, and burnout [8,9], particularly in anaesthesia, intensive care, and emergency room staff [10]. Several meta-analyses confirmed high levels of anxiety, distress, and depression in front-line HCPs [11,12,13,14,15]. These meta-analyses included short-term cross-sectional studies comparing psychological symptoms to non-pandemic values. The absence of longitudinal studies made it difficult to distinguish the effect of the pandemic from other work-related stressors [16].

Work-related stress, anxiety, and depression impair HCPs’ wellbeing and their ability to work [17] and might contribute to reduced healthcare quality, with poor patient care affecting patient outcomes [18]. Work-related Sense of Coherence (Work-SoC), an adaptive dispositional orientation that enables coping with adverse experiences, often acts as a psychosocial protection against such mental strain [19] and is considered an indicator for the health-promoting quality of life at work [20] and work-related wellbeing [21].

Work-SoC is a salutogenic concept and indicates the perceived quality of employees’ health-relevant work conditions. It is sensitive to changes in the work environment and is conceptualised with three dimensions: (1) comprehensibility, the cognitive component describing the work situation as structured, consistent, and clear [19]; (2) manageability, the instrumental component describing the extent to which the individual perceives the resources that are available to cope with demands posed by the work environment; and (3) meaningfulness, the motivational component reflecting whether the work situation is seen as worthy of commitment and involvement [22].

Some authors have reported that the comprehensibility component was demonstrated to have the highest correlations with most working conditions and health outcomes in contrast to the other components in the general population under non-pandemic conditions [23]. In recent studies of HCPs, the comprehensibility component of general SOC appeared to be the most efficient protector against psychological distress, followed by manageability [24,25]. Antonovsky himself assumed a strong relation between the manageability and comprehensibility components in the context of work [26]. The manageability component, describing a feeling of having control over the demands of the environment, plays an important role in connection with the appearance of post-traumatic and depressive symptoms [26,27]. Further, the manageability and meaningfulness components of general SOC seem to be important resources for nurses, buffering the negative impact of mental load on professional burnout [28], whereas the meaningfulness component, which is defined as experiencing the environment as significant and which reflects perceived learning and development opportunities, appeared to have the highest predictive value for professional burnout in HCPs during the pandemic [26,29].

Little is known about the influence of Work-SoC on the psychological health of acute care HCPs working during the COVID-19 pandemic. The aim of our study was to evaluate the relationship between Work-SoC and variations in the psychological health of anaesthetists and emergency care and critical care physicians, as demonstrated by changes in COVID-19-related anxiety, perceived vulnerability, depressiveness, and symptoms of psychological trauma during the first year of the pandemic, and how these differ between front-line and second-line HCPs.

## 2. Materials and Methods

The detailed study protocol with the methods and procedure was published previously [30]. In brief: After ethical waiver, a link to the survey was sent to possible study participants. Those who participated consented electronically and provided demographic data. Data regarding the HCPs’ mental health during the COVID-19 pandemic were retrieved in April 2020 (baseline—T0), July 2020 (T1), October 2020 (T2), January 2021 (T3), and April 2021 (T4). The survey link to the questionnaire was distributed using the ‘snowballing’ sampling technique. The survey consisted of six validated self-reporting questionnaires assessing the HCPs’ psychological health [30]. We included HCPs [30] older than 18 years who either had direct contact (i.e., diagnosed, treated, or provided care) with COVID-19-infected patients (medical front-liners) or did not (medical second-liners).

The primary study outcome was COVID-19-related anxiety, measured with the modified Swine Influenza Anxiety Index (m-SFI) [30]. The secondary outcomes included: individual beliefs concerning susceptibility to infectious diseases and transmission of pathogens, measured by the Perceived Vulnerability to Disease Scale (PVD) [31]; depressiveness, assessed with the Patient Health Questionnaire (PHQ-9) [32]; and trauma symptomatology, assessed with the Impact of Event Scale-6 (IES-6) [33]. All these outcomes were compared with an individual’s Work-SoC assessed by the work-related Sense of Coherence Scale [19]. A higher score represents a higher Work-SoC. The questionnaire was hosted online at Qualtrics (Provo, UT, USA), limiting access to one response per device.

### Statistical Analysis

A minimum sample size of *n* = 69 resulted from an a priori power analysis for a repeated-measure analysis of variance with five time points, an α error of 0.05, and a β of 80%. R statistical language (R Foundation for Statistical Computing, Vienna, Austria) with the packages nlme, reghelper, emmeans, and DHARMa was used [34]. A multilevel logistic regression analysis accounted for the hierarchical data structure (time points at level 1 are nested within individuals at level 2) [35]. Continuous predictors were mean-centred to reduce multicollinearity. We applied restricted maximum likelihood (REML) for parameter estimation to minimise bias in estimates of variance and covariance parameters. The normal distributions of the outcome variables were examined by inspecting residual diagnostics of the fitted multilevel models.

We calculated four different multilevel models for each outcome variable. The null intercept-only model (without predictors) was estimated for the inter-correlation coefficient (ICC). The ICC represents the estimated proportion of variation in the outcome variables to determine whether a three-level model with participants grouped to different world regions [36] as the third level significantly improves the model fit. For model 1 (nonlinear unconditional growth model with random intercept), we explored the within-participant trajectories of the cubic change across the measurement points. The Work-SoC level effect on the outcome across different time point analyses included a conditional growth model with random intercept and cross-level interaction (model 2) with all predictor variables and a two-way cross-level interaction of Work-SoC and time point. In model 3, a conditional growth model with three-way cross-level interactions, we examined the three-way cross-level interaction of Work-SoC levels, time points, and front-line and second-line HCPs. Including random intercepts and slopes improved all of the models significantly. We inspected the Akaike Information Criterion (AIC) and the Bayesian Information Criterion (BIC) to compare the different models [35]. The a priori level of significance was 0.05 (2-sided).

## 3. Results

Two-thirds of our sample were anaesthesia, critical care, and emergency care staff. At the baseline measurement (April 2020), 1578 participants responded. Five hundred and twenty participants who completed at least four out of five surveys were included (response rate 33.0%, Figure 1). Most participants were female (62%, *n* = 322), aged 41.6 ± 10.7 years, front-line HCPs (95%, *n* = 492), and European (Western, *n* = 258; 49.6%; Southern, *n* = 112; 21.5%; Northern, *n* = 80, 15.4%). Table 1 displays the participants’ characteristics and data from non-completers addressing a potential attrition bias.

### 3.1. Primary Outcome: Predictors of COVID-19-related Anxiety

Model 1 for COVID-19-related anxiety revealed a significantly improved fit for the inclusion of a cubic term (AIC = 14,022.3; BIC = 14,149.2, *p* = 0.001), with a significant relationship between time and COVID-19-related anxiety (Appendix A). Model 2 showed a significant negative relationship between the time–Work-SoC interaction and COVID-19-related anxiety (*b* = 0.01, SE = 0.002, *p* < 0.001). The slopes are shown in Figure 2a. Post hoc, COVID-19-related anxiety trajectories differed between high- and low-level Work-SoC HCPs (*p* = 0.002). HCPs who reported close contact with COVID-19 risk groups showed higher degrees of COVID-19-related anxiety than those who did not (*b* = 1.95, SE = 0.689, *p* = 0.006). Front-line and second-line HCP status was not a significant predictor of COVID-19-related anxiety (*p* = 0.161). The model explained 52.7% of the variance. In model 3, the added three-way interaction of front-line and second-line HCPs, Work-SoC, and time point was not significant (*p* = 0.771) (Appendix A), with no significant moderating effects of front-line or second-line HCPs on the Work-SoC time interaction.

### 3.2. Secondary Outcomes

#### 3.2.1. Predictors of Perceived Vulnerability to Disease (PVD)

Model 1 for perceived vulnerability to disease showed an improved fit for the inclusion of a “u-shaped” term (AIC = 18232; BIC = 18289.7, *p* < 0.001), with no significant relationship between time and PVD (Appendix A). However, the “u-shaped” cubic trajectory of PVD was influenced by Work-SoC (model 2, *b* = 0.01, SE = 0.005, *p* = 0.004). The Work-SoC slopes are shown in Figure 2b. Post hoc, all slopes differed (*p* = 0.003). Moreover, a younger age (*b* = −0.125, SE = 0.051, *p* = 0.014) and not belonging to a COVID-19 risk group (*b* = −3.02, SE = 1.389, *p* = 0.030) were associated with a lower PVD. Being a front-line or second-line HCP was not a predictor of PVD (*p* = 0.245). The model explained 72.1% of the variance. The three-way interaction of front-line and second-line HCPs, Work-SoC, and time point showed no differences in the trajectories of PVD between the front-line and second-line HCPs (*p* = 0.097; Appendix A, model 3).

#### 3.2.2. Predictors of Depressiveness

Model 1 for depressiveness showed improved fit for the inclusion of a “u-shaped” trajectory (AIC = 13,743.1; BIC = 13,800.8, *p* < 0.001), with a relationship between time and depressiveness (Appendix A). The relationship between the time–Work-SoC interaction and depressiveness indicated that Work-SoC influenced the cubic trajectory (model 2, *b* = 0.01, SE = 0.000, *p* < 0.001). The Work-SoC slopes are shown in Figure 2c. Post hoc, all slopes differed (*p* = 0.001). Being male (*b* = 1.05, SE = 0.31, *p* < 0.001) and not belonging to the COVID-19 risk group (*b* = −1.30, SE = 0.451, *p* = 0.004) were associated with lower depressiveness. Participants with an uncertain COVID-19 infection status (*b* = 0.210, SE = 0.077, *p* = 0.006) reported more symptoms of depression compared to other participants (infected or not). Being a front-line or second-line HCP was not a predictor of depressiveness (*p* = 0.766). The model explained 59.1% of the variance. The added three-way interaction of front-line and second-line HCPs, Work-SoC, and time point indicated no differences in the trajectories of depressiveness between the front-line and second-line HCPs (*p* = 0.956; Appendix A Appendix A, model 3).

#### 3.2.3. Predictors of Psychological Trauma Symptomatology

Model 1 for psychological trauma symptomatology showed an improved fit for the inclusion of a cubic term (AIC = 14,131.4; BIC = 14,189.1, *p* < 0.001), with no significant relationship between time and psychological trauma symptomatology (Appendix A). The relationship between the time–Work-SoC interaction and psychological trauma symptomatology indicated that Work-SoC influenced the cubic trajectory (model 2, *b* = 0.012, SE = 0.002, *p* < 0.001). The Work-SoC slopes are shown in Figure 2d. Post hoc, all slopes differed (*p* < 0.001). A younger age (*b* = −0.039, SE = 0.017, *p* = 0.025) and the uncertainty of a COVID-19 infection (*b* = −0.195, SE = 0.080, *p* = 0.016) was associated with higher psychological trauma symptomatology. Being a front-line or second-line HCP was not a significant predictor of psychological trauma symptomatology (*p* = 0.538). The model explained 54.3% of the variance. The added three-way interaction of front-line and second-line HCPs, Work-SoC, and time point was not a predictor of psychological trauma symptomatology (*p* = 0.757; Appendix A, model 3), which indicated no differences in the trajectories of psychological trauma symptomatology between the front-line and second-line HCPs.

## 4. Discussion

This one-year longitudinal observational study analysed the relationship between Work-related Sense of Coherence (Work-SoC) and the psychological health of anaesthesia, emergency care, and critical care staff. Different levels of Work-SoC were associated with different levels of COVID-19-related anxiety, perceived vulnerability to disease (PVD), and symptoms of depressiveness and psychological trauma. Not belonging to the COVID-19 risk group was a protective factor for depressiveness and PVD. COVID-19-related anxiety and depressiveness increased with time. The trajectories for depressiveness and psychological trauma symptoms showed an association with Work-SoC in the early stages of the pandemic, but this effect was neutralised with time. There was no difference in psychological health patterns between front-line and second-line HCPs. 

Sense of Coherence (SoC), conceptualised as a global orientation viewing life as structured, manageable, and meaningful [26,37,38], seems to be an essential aspect of understanding individuals coping with stress [39]. SoC is influenced by life experiences outside work [37]. Consequently, we chose a more context-specific definition of SoC, the Work-related SoC, that is more sensitive to changing working conditions [19]. Studies have shown a positive correlation between Work-SoC and mental health, work enthusiasm, job resources, and effective organisational commitment [19,23]. Additionally, compared to the global SoC, Work-SoC is a better predictor for work engagement [40].

While Work-SoC is a novel concept and evidence is scarce, a vast body of literature supports the positive relationship between SoC, health, and quality of life during stressful events, across both life stages and cultures [39,41,42,43]. During the COVID-19 pandemic, studies identified SoC as a unique resource [44,45]. Barni et al. [44] demonstrated a direct positive association between SoC and wellbeing. Schäfer and colleagues pointed out that SoC could predict changes in psychological symptoms during the pandemic’s evolution and could act as a “buffer” for clinically relevant symptoms of depression in individuals with higher SoC levels [45]. These findings support our results, as in our sample, individuals with higher levels of Work-SoC consistently showed fewer psychological symptoms throughout the pandemic. Although it was initially hypothesised that SoC could plateau during early adulthood [26,38,46], research prior to the pandemic has shown that SoC increases slightly with age and decreases during adverse life events [47]. Our results again support this, as individual levels of Work-SoC decreased in all studied outcomes. 

The fluctuations in the trajectory of COVID-19-related anxiety levels for HCPs with different levels of Work-SoC was puzzling. HCPs with lower Work-SoC levels (–1SD) reported decreased COVID-19-related anxiety from April 2020 to January 2021, but this then slightly increased to April 2021. HCPs with high degrees of Work-SoC (+1SD) followed a similar pattern, although their lowest degree of anxiety was in October 2020, and this then increased steeply to April 2021. This might be interpreted as indicating that those individuals with a lower Work-SoC are able to recover faster to baseline levels, a phenomenon which has not been described previously. 

Given that anxiety may be viewed as a sequela of maladaptive coping with stress, one would expect Work-SoC and anxiety to have inversely related trajectories. Instead, the proportion of staff reporting anxiety did not continue to increase after early lockdown phases in 2020, suggesting that HCPs had ways to cope that allowed their anxiety to return to earlier levels after pandemic-response events. We hypothesise that the “u-shaped” pattern seen in COVID-19-related anxiety may reflect variations due to entering and exiting lockdowns [48,49], available vaccinations, or other unknown individual or contextual factors interacting with Work-SoC. As low-Work-SoC individuals experienced more significant COVID-19-related anxiety initially, they might show more improvement in their psychological symptoms over time [50], which would be a ceiling effect for low-Work-SoC HCPs. 

PVD levels varied less over time and averaged out in April 2021. Previous studies during the pandemic showed that PVD: (1) had shifted significantly as a function of the pandemic [51,52], (2) is prone to experimental manipulation (participants who read the coronavirus morbidity–mortality statistics scored higher on PVD compared to those who did not read such information [53]), and (3) predicts preventive behaviours [54,55,56]. Our sample was a cohort of anaesthesia, emergency care, and critical care staff, considered experts and assumed to have better or more extensive disease knowledge, and with high motivation for preventive behaviours (personal protective equipment, vaccinations). Consequently, we expected higher levels of PVD in our sample compared to the general population, which may explain why variations were not so apparent.

Being a woman and belonging to a COVID-19 risk group were risk factors for symptoms of depressiveness. This aligns with previous findings: women have a higher lifetime prevalence for developing mental disorders [57] and showed a lower level of SoC during the COVID-19 pandemic [46]. A younger age was also a previously described risk factor [46,58]. 

The trajectories of the symptoms of depressiveness and psychological trauma symptomatology were very similar. In April 2020, participants with a higher level of Work-SoC showed fewer symptoms of depressiveness and psychological trauma symptomatology than participants with an average or a lower level, which reflects the Work-SoC’s wellbeing component. During the course of the study, these participants (+1SD) developed more symptoms of depressiveness and psychological trauma symptomatology. The average level of Work-SoC was associated with a u-shaped curve, with HCPs with lower Work-SoC levels (–1SD) developing fewer symptoms of depressiveness and psychological trauma symptomatology. This suggests that a higher level of Work-SoC is protective against developing symptoms of depression and psychological trauma symptomatology, but only in the early stages of a pandemic (Figure 2c,d). The depletion of resources of participants with a higher level of Work-SoC might be caused by continuously higher job demands during the pandemic without the prospect of an end. According to Vogt et al., [19] higher Work-SoC levels were related to a higher draw on resources from work, whereas lower levels of Work-SoC comprised higher work demands. Nevertheless, over time, higher levels of Work-SoC might be a risk factor for developing symptoms of depression and psychological trauma symptomatology, while lower levels of Work-SoC might be protective. This needs to be confirmed with further studies.

Our models found no differences between first-line and second-line HCPs’ wellbeing during the COVID-19 pandemic, despite yielding a significant proportion of explained variance. This is in contrast to previously published reports that HPCs with a higher degree of exposure to suspected/confirmed cases have poorer psychological outcomes than those unexposed [5,6,8,9]. We hypothesise that external societal stressors influenced both types of HCP identically; work-related wellbeing was already degraded previously to the pandemic, and therefore our results are skewed, making differences less statistically relevant (though we did not investigate burnout and have no pre-pandemic data); most HCPs in our sample did not have to work in environments unknown to them (only a small, non-significant fraction was pulled out of their usual working environment to a new one), and surprisingly, job satisfaction during the COVID-19 crisis has stayed high in several subgroups of HCPs [59,60,61,62]. Staff perceptions of the value, impact, and contribution of their work to society during the COVID-19 crisis have been shown to contribute to high levels of professional satisfaction, and the unfavourable working conditions [60,61] and lack of equipment and human resources had little effect on professional satisfaction [59]. This paradox could be explained by the fact that dissatisfaction regarding the lack of equipment or human resources was directed toward the work organisation and the political management of the crisis. Professional satisfaction remained high because it is related to the workers’ value and role in helping others.

A limitation of our study is a possible response bias (i.e., social desirability) and considerable regional variations in the COVID-19 pandemic, although this did not interfere with the statistical models. We included only baseline scores of Work-SoC to keep the complexity low and avoid overloading the models, but we were unable to obtain pre-pandemic Work-SoC levels. Therefore, we cannot exclude that the pre-existing mental diagnoses of the HCPs influenced our results [50]. Finally, the study sample contained HCPs from all over the globe. The COVID-19 evolution at the measured time points may have been distinct in different regions, undermining our results. However, we were mainly interested in understanding the association between Work-SoC and psychological health during the pandemic and how it operated beyond specific contexts or cultures. We therefore assumed the potential universality of the positive effect of Work-SoC in times of a global crisis. Based on Antonovsky’s assumption about the universal SoC [26,38], we expected to find similar patterns among people from different countries. Finally, while we considered the snowball sampling method as the most feasible method for this study, we have to acknowledge its limitations: because the researcher has little control over the sampling, it may lead to unintentionally overvaluing some groups in comparison to others, so the representativeness of the sample is not guaranteed.

## 5. Conclusions

So far, only a few studies have investigated the sense of coherence during the COVID-19 pandemic, and they are either cross-sectional [29,44,63,64,65,66] or limited to a single geographical area [46,65]. This is the first time that a prospective longitudinal observational study has evaluated the trajectories of mental health in healthcare workers using the more sensitive Work-related Sense of Coherence. Our results bring new insights into the psychological impacts on anaesthesia, emergency care, and critical care staff during the first year of the COVID-19 pandemic. The degree of COVID-19-related anxiety and perceived vulnerability to disease were elevated among HCPs regardless of their amount of contact with COVID-19 patients, while the trajectories for HCPs with high and low Work-SoC (±1SD) differed significantly. In contrast to previous studies, front-line and second-line HPCs had different trajectories of depressiveness and psychological trauma symptomatology. Second-line HCPs with high and average degrees of sense of coherence showed a steeper worsening of their depressiveness compared to front-line HCPs with high and average degrees of sense of coherence. Establishing a clear relationship between psychological symptomatology and work-related sense of coherence with the development of mental symptoms during exceptional situations such as the current COVID-19 pandemic helps to identify HCPs who are both particularly protected and at risk, which will allow the adequate distribution of psychological interventions. Organisations can also potentiate Work-SoC in their employees by ensuring that they are adequately trained. This is would be an affordable measure that could save money and resources by keeping staff at work and avoiding sick leave. Finally, the specific resources identified might be used to buffer the long-term effects of increased demands on anaesthesia, emergency care, and critical care staff during a pandemic.

## Figures and Tables

**Figure 1 ijerph-19-06053-f001:**
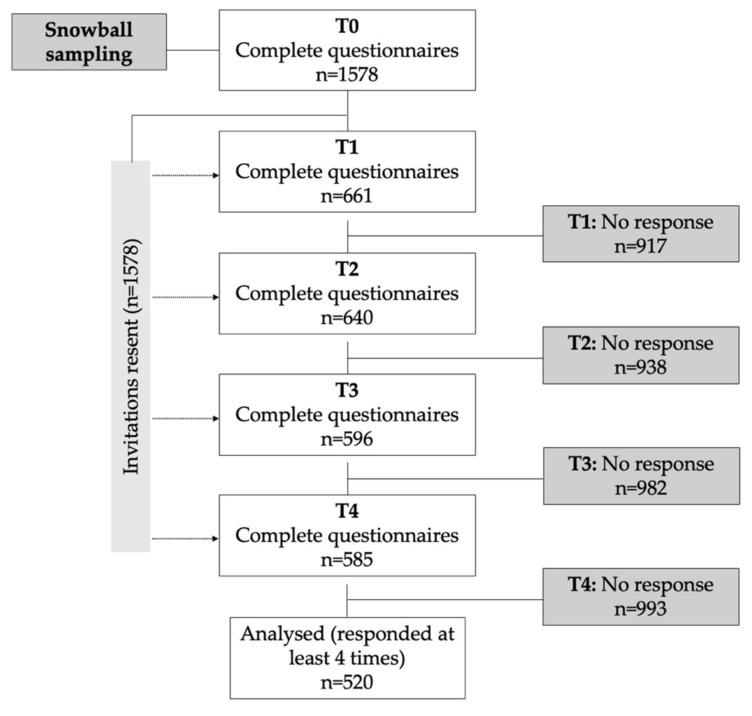
Study Flowchart. April 2020 (baseline—T0), July 2020 (T1), October 2020 (T2), January 2021 (T3), and April 2021 (T4).

**Figure 2 ijerph-19-06053-f002:**
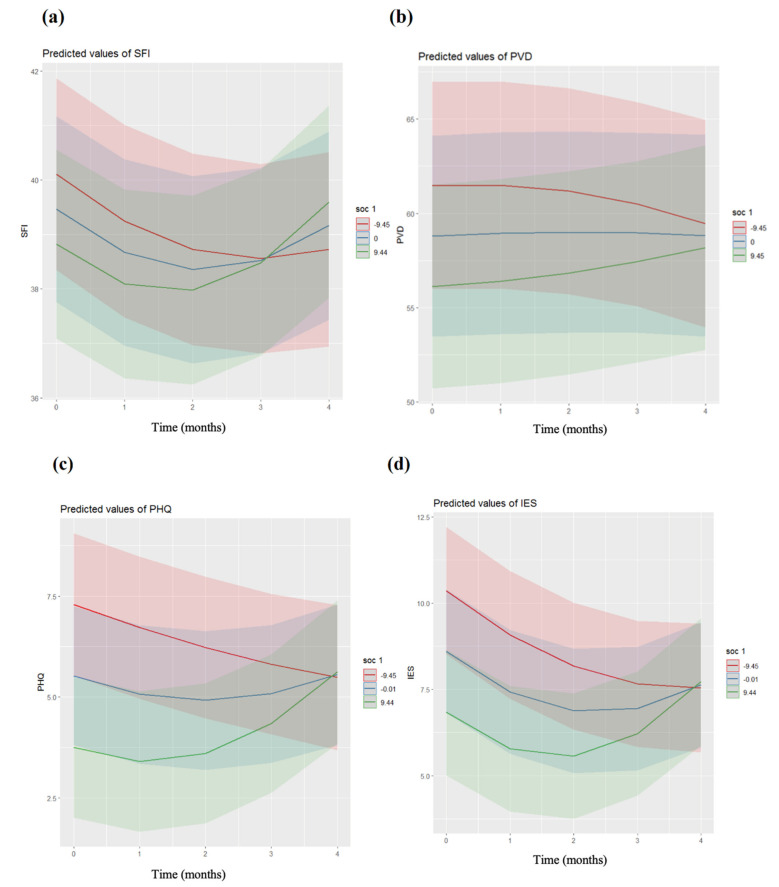
Trajectories of the mental health of the healthcare professionals across the measurement points for their Work-SOC levels (red, –1SD; blue, mean; green, +1SD). Slopes of COVID-19-related anxiety (**a**), perceived vulnerability (**b**), depressiveness (**c**), and psychological trauma symptomatology (**d**), all by Work-SOC levels. The shaded areas correspond to the confidence intervals of each trajectory. Number −9.45 = −1SD (lower Work-SoC), +9.45 = +1SD (higher Work-SoC). Time points: T0, April 2020; T1, July 2020; T2, October 2020; T3, January 2021; T4, April 2021. m-SFI, modified Swine Influenza Anxiety Index; PVD, Perceived Vulnerability to Disease Scale; PHQ, Patient Health Questionnaire; IES, Impact of Event Scale; Work-SoC, Work-related Sense of Coherence Scale. Colour-shaded areas represent the SD. The figures were directly plotted by R package.

**Table 1 ijerph-19-06053-t001:** Characteristics of the healthcare professionals included in this study.

Characteristic	Completers *	Non-Completers	Statistics
			*p* ^†^
Total	520	1058	
Age (years)			
Mean (SD)	41.55 (10.69)	40.11 (10.38)	0.01
Gender, n (%)			0.90
Female	321 (61.7)	661 (62.5)	
Male	198 (38.1)	394 (37.2)	
Other	1 (0.2)	3 (0.3)	
Risk group, n (%)			0.20
No	454 (87.3)	897 (84.8)	
Yes	66 (12.7)	161 (15.2)	
Occupation, n (%)			0.01
Nurse	88 (16.9)	243 (23.0)	
Physician	364 (70.0)	667 (63.0)	
Other	68 (13.1)	148 (14.0)	
Workplace, n (%)			0.39
ICU	117 (22.5)	269 (25.4)	
Anaesthesia	184 (35.4)	396 (37.4)	
Emergency room	37 (7.1)	71 (6.7)	
Ward	38 (7.3)	63 (6.0)	
Other	144 (27.7)	259 (24.5)	
Work status, n (%)			0.78
Front-line	334 (64.2)	688 (65.0)	
Second-line	186 (35.8)	370 (35.0)	
Contact with COVID-19 patients during study, n (%)			
No	28 (5.4)	436 (41.2)	0.00
Yes	492 (94.6)	622 (58.8)	
In a relationship, n (%)			0.89
No	81 (15.6)	162 (15.3)	
Yes	439 (84.4)	896 (84.7)	
Household, n (%)			0.21
Live alone	78 (15.0)	185 (17.5)	
Live with someone	442 (85.0)	873 (82.5)	
Children, n (%)			0.42
No	240 (46.2)	511 (48.3)	
Yes	280 (53.8)	547 (51.7)	
Infected with COVID-19 during study, n (%)			0.00
No	232 (44.6)	599 (56.6)	
Yes	67 (12.9)	32 (3.00)	
Do not know	221 (42.5)	427 (40.4)	
Contact with COVID-19 patients during study, n (%)			0.00
No	28 (5.4)	436 (41.2)	
Yes	492 (94.6)	622 (58.8)	
World region, n (%)			0.02
Western Europe	258 (49.6)	450 (42.5)	
Southern Europe	112 (21.5)	266 (25.1)	
Northern Europe	80 (15.4)	145 (13.7)	
North America	43 (8.3)	113 (10.7)	
Other regions	27 (5.2)	84 (7.9)	
Sense of Coherence (Work-SOC), n (%)			
Low (−1SD)	94 (18.1)		
Average	335 (64.4)		
High (+1SD)	91 (17.5)		

* Participants with a minimum of four out of five measurement points. ^†^ Comparisons between completers and non-completers (two-sided Welch’s *t*-tests for continuous data; Pearson’s chi-squared tests for categorical variables). Work-SOC, work-related Sense of Coherence Scale; ICU, intensive care unit; *p*-value; SD, standard deviation.

## Data Availability

The data that support the findings of this study are available with a research question and ethics approval on request from the corresponding author. The data are not publicly available due to privacy or Swiss legal restrictions.

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
