# Peer review of "Health-Promoting Quality of Life at Work during the COVID-19 Pandemic: A 12-Month Longitudinal Study on the Work-Related Sense of Coherence in Acute Care Healthcare Professionals"

_ijerph, 2022, doi:10.3390/ijerph19106053_

Round 1

Reviewer 1 Report

The paper assesses individual dimensions of QoL of healthcare personnel during COVID-19 pandemic. I consider the paper to be of high quality, well-founded, and of top interest for readers. However, I recommend some changes to make the paper even more attractive:

  1. The recruitment method had necessarily led to a selection bias. Although it is mentioned in the Discussion, it should be discussed in more depth. In my opinion, the chosen method was adequate in the COVID times; it was impossible to arrange for a better selection scheme. Nevertheless, the convenience (snow-ball) sampling leads to overvaluing of some groups while undervaluing of others, as Table 1 shows. Please discuss it in more detail (the results are not generally valid for all healthcare workers, but only for the included strata).
  2. Figure 1 is not enough informative. It should show the decrease of respondents in each round, and the composition of the final sample. Try to find another presentation of the figures.
  3. In Table 1, the headings of three columns of "Statistics" are not fully clear. Please make clear what each column means (moreover, there is something missing in the second column denoted only "(df)").

Author Response

Comment 1: The paper assesses individual dimensions of QoL of healthcare personnel during COVID-19 pandemic. I consider the paper to be of high quality, well-founded, and of top interest for readers. However, I recommend some changes to make the paper even more attractive.

Reply: Thank you. Your comments to improve our manuscript are much appreciated.

 Comment 2: The recruitment method had necessarily led to a selection bias. Although it is mentioned in the Discussion, it should be discussed in more depth. In my opinion, the chosen method was adequate in the COVID times; it was impossible to arrange for a better selection scheme. Nevertheless, the convenience (snow-ball) sampling leads to overvaluing of some groups while undervaluing of others, as Table 1 shows. Please discuss it in more detail (the results are not generally valid for all healthcare workers, but only for the included strata).

Reply: Thank you. We agree with your comment and have extended the limitations section further. You can now read (Page 13, Lines 334-338): “Finally, while we considered the snowball sampling method as the most feasible method for this study, we have to acknowledge its limitations: because the researcher has little control over the sampling, it may lead to unintentionally overvaluing some groups in comparison to others, so representativeness of the sample is not guaranteed.”

Comment 3: Figure 1 is not enough informative. It should show the decrease of respondents in each round, and the composition of the final sample. Try to find another presentation of the figures.

Reply: Thank you for this important remark. We have changed Figure 1 accordingly. Please note that all initial participants were contacted in all four rounds and we accepted participants who responded at least in four of the five time-points. We therefore did not exclude participants that had not replied to the survey in previous rounds. We hope that this is now clear on the current flowchart.

Comment 4: In Table 1, the headings of three columns of "Statistics" are not fully clear. Please make clear what each column means (moreover, there is something missing in the second column denoted only "(df)").

Reply: Thank you. We realise that the three columns bring more confusion than information so we have decided to leave the p-value only.

Reviewer 2 Report

The submitted work by Berger-Estilita and colleagues demonstrated the impact of COVID-related situations on the well-being of healthcare professionals in acute care settings. The authors conducted an online survey at different time points. They analysed the data using a hierarchical modelling approach, by proposing three models and a null model. One primary outcome measure (COVID-related anxiety) and 3 secondary outcome measures (vulnerability to disease, depressiveness, psychological trauma) were examined with several predictor variables (demographic, time, Work-SoC, etc) in the analysis. It was found that Work-SoC was a good predictor for the outcome measures which bears some importance, especially in the public health sense. The authors included the STROBE checklist and the full statistical results as Supplementary materials.

I wish to clarify some things:

  1. Perhaps the authors can mention the survey platform used (Qualtrics, Google Forms etc). If this is an anonymous survey, the authors can state so.
  2. Did the authors consider a simpler model without the 2-way or 3-way cross-level interactions? An overly-complex model may complicate the interpretation. Also, I wonder if the authors meant “quadratic” instead of “cubic” if the variable Time has only power^2.
  3. Figure-2 is rather confusing and the image quality is quite poor (I understand this is fresh from the ggplot). If the 3 different colours refer to the mean and +/- 1 SD of Work-SoC, what are the shaded areas? And what are the number -9.45 and +9.44? Can the authors reproduce Figure-2 to have a sharper image without being cropped on the left side?
  4. Line 109-112: the Methods mentioned using three-level modelling (with region or place as the highest level), but was this used or reported in the Results?

Author Response

Comment 1: The submitted work by Berger-Estilita and colleagues demonstrated the impact of COVID-related situations on the well-being of healthcare professionals in acute care settings. The authors conducted an online survey at different time points. They analysed the data using a hierarchical modelling approach, by proposing three models and a null model. One primary outcome measure (COVID-related anxiety) and 3 secondary outcome measures (vulnerability to disease, depressiveness, psychological trauma) were examined with several predictor variables (demographic, time, Work-SoC, etc) in the analysis. It was found that Work-SoC was a good predictor for the outcome measures which bears some importance, especially in the public health sense. The authors included the STROBE checklist and the full statistical results as Supplementary materials.

Reply: Thank you for this nice summary. Your comments to improve our manuscript are much appreciated.

Comment 2: I wish to clarify some things: Perhaps the authors can mention the survey platform used (Qualtrics, Google Forms etc). If this is an anonymous survey, the authors can state so.

Reply: Thank you for this comment. We apologise for this omission. We have duly added the survey platform (Page 2, Lines 90-92): “The questionnaire was hosted online at Qualtrics (Provo, Utah, USA), which restricts access to one response per device.”

Comment 3: Did the authors consider a simpler model without the 2-way or 3-way cross-level interactions? An overly-complex model may complicate the interpretation. Also, I wonder if the authors meant “quadratic” instead of “cubic” if the variable Time has only power^2.

Reply: Thank you for this important remark. We have indeed considered a hierarchical approach. First, the unconditional model (more simple) presented in each table, and second, the unreported models with no interactions. For all outcomes, the model fits were significantly better (AIC, BIC) when including a reported 2-way or 3-way interaction. Cubic for non-linear gradients, quadratic for linear gradients. We are glad that you spotted this typo and have now adjusted the tables accordingly, substituting all  ^2 with ^3 in the Supplementary File. We hope that this point is clearer now.

Comment 4: Figure-2 is rather confusing and the image quality is quite poor (I understand this is fresh from the ggplot). If the 3 different colours refer to the mean and +/- 1 SD of Work-SoC, what are the shaded areas? And what are the number -9.45 and +9.44? Can the authors reproduce Figure-2 to have a sharper image without being cropped on the left side?

Reply: Thank you for this remark. We have reproduced the image in better quality and integrity, as suggested. We also added in the legend the meaning for the shaded areas and the numbers. Shaded areas are the Confidence Intervals of each trajectory. Number -9.45 = - 1SD (lower Work-SoC), +9.45 = +1SD (higher Work-SoC). The figures are directly plotted by the software R package plot and unfortunately, no modifications are allowed.

Comment 5: Line 109-112: the Methods mentioned using three-level modelling (with region or place as the highest level), but was this used or reported in the Results?

 Reply: Thank you. We apologise if our description of the results was more ambiguous than intended. We mentioned that we would “determine whether a three-level model with participants grouped to different world regions as the third level significantly improved the model fit” (Page 3, lines 124-125). The world regions served as a “control” and not a predictor. Since we could not find any differences in the model fit, we did not report it in the Results section.

Reviewer 3 Report

The aim of reviewed article is to “evaluate Work-SoC with variations of psychological health of anaesthetists, emergency care and critical care physicians demonstrated by changes in COVID-19–related anxiety, perceived vulnerability, depressiveness, and symptoms of psychological trauma during the first year of the pandemic, and how these differ between front-line and second-line HCPs”.

The article brings sufficient contribution to the development of knowledge in the concerned area. The article closely corresponds to the topic specified in the title. In this article all the issues were discussed in an understandable manner.

I suggest supplementing the subject literature and expanding the introduction. I recommend extending the conclusions. Concluding remarks should be clear and comprehensive.

I recommend checking certain aspects in the bibliographic reference lists. For example, the name of the journal must be written with its abbreviation and the volume must be written in italics. It requires improvements.

Author Response

Comment 1: The aim of reviewed article is to “evaluate Work-SoC with variations of psychological health of anaesthetists, emergency care and critical care physicians demonstrated by changes in COVID-19–related anxiety, perceived vulnerability, depressiveness, and symptoms of psychological trauma during the first year of the pandemic, and how these differ between front-line and second-line HCPs”.

The article brings sufficient contribution to the development of knowledge in the concerned area. The article closely corresponds to the topic specified in the title. In this article all the issues were discussed in an understandable manner.

Reply: Thank you for your kind summary.

Comment 2: I suggest supplementing the subject literature and expanding the introduction.

Reply: Thank you. Due to word count constraints, we were only able to introduce the following paragraphs. (Page 2, lines 67-81) “Bauer, Vogt, Inauen and Jenny reported that the comprehensibility component was demonstrated to have the highest correlations with most working conditions and health outcomes in contrast to the other components in the general population under non-pandemic conditions. In recent studies of HCPs, the comprehensibility component of general SOC appeared to be the most efficient protector against psychological distress, followed by manageability. Antonovsky himself assumed a strong relation between the manageability and comprehensibility components in the context of work. The manageability component, describing a feeling of having control over the demands of the environment, plays an important role in connection with the appearance of posttraumatic and depressive symptoms. Further, manageability and meaningfulness of general SOC seem to be important resources for nurses, buffering the negative impact of mental load on professional burnout, whereas the meaningfulness component, defined by experiencing the environment as significant and which reflects perceived learning and development opportunities, appeared to have the highest predictive value for professional burnout in HCPs during a pandemic.” We hope this expansion may accommodate your request.

Comment 3: I recommend extending the conclusions. Concluding remarks should be clear and comprehensive.

Reply: Thank you for this critical remark. We have extended the conclusions accordingly and now you can read: (Page 11, Lines 349-365) “The degree of COVID-19–related anxiety and perceived vulnerability to disease were elevated among HCPs regardless of their amount of contact with COVID-19 patients, while the trajectories for HCPs with high and low Work-SoC (±1SD) differed significantly. In contrast to previous studies, front-line and second-line HPCs had different trajectories of depressiveness and psychological trauma symptomatology. Second-line HCPs with high and average degrees of Work-SoC showed a steeper worsening of their depressiveness compared to front-line HCPs with high and average degrees of Work-SoC. Establishing a clear relationship between Work-SoC and the development of mental symptoms during exceptional situations (i.e. the current COVID-19 pandemic) helps to identify HCPs who are both particularly protected and at risk. This will allow the need-based distribution of psychological interventions. Further investigations are needed on if and how to improve HCPs’ Work-SoC. This would be an affordable measure that can save money and resources by keeping the staff at work and avoiding sick leave. Finally, the identified specific resources might be used to buffer long-term effects of increased demands in acute care HCPs during a pandemic.”

Comment 4: I recommend checking certain aspects in the bibliographic reference lists. For example, the name of the journal must be written with its abbreviation and the volume must be written in italics. It requires improvements.

Reply: Thank you for pointing this out to us. We have checked the journal guidelines and changed the references accordingly.